# A New Method for Analyzing the Aero-Optical Effects of Hypersonic Vehicles Based on a Microscopic Mechanism

Bo Yang [1], He Yu [1,*], Chaofan Liu [1], Xiang Wei [1], Zichen Fan [2] and Jun Miao [3]

[1] School of Astronautics, Beihang University, Beijing 100191, China
[2] Beijing Institute of Control and Electronic Technology, Beijing 100038, China
[3] Qian Xuesen Laboratory of Space Technology, Beijing 100094, China
* Correspondence: 13261059095@163.com

**Abstract:** Aero-optical effects are the key factors that restrict the accuracy of the optical sensors of hypersonic vehicles, and the numerical simulation of aero-optical effects is a powerful tool with which to analyze aero-optical distortion. Most existing research focuses on the simulation analysis of refraction distortion based on the density field at the macro level via the ray-tracing method. In this paper, a method for analyzing aero-optical effects based on the interaction between photons and gas molecules is proposed and can explain the optical distortion and energy dissipation caused by aero-optical effects at the micro level. By establishing a transmission model of photons in turbulence, a simulation method of aero-optical effects based on a microscopic mechanism is designed and breaks through the limitations of a traditional macro method in energy analyses. The optical distortion parameters based on photonics are compared with the physical quantities of traditional aero-optical effects, which verifies the effectiveness of the micro analysis on the macro scale and provides a new idea for studying the microscopic mechanism of aero-optical effects.

**Keywords:** hypersonic vehicles; aero-optical effects; photonics; optical sensors; numerical simulation

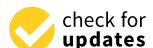



## 1. Introduction

Hypersonic vehicles have received a significant amount of attention due to their unique ability to rapidly attack [1,2]. With the increasing military demand for them, autonomous navigation technology has become an important challenge for the further development of hypersonic vehicles [3]. Compared with other navigation systems, a celestial navigation system (CNS) has unparalleled advantages, such as non-cumulative error, strong anti-interference ability, and high reliability [4,5]. However, the measurement errors of optical sensors caused by aero-optical effects are a key factor affecting the application of CNSs in hypersonic vehicles [6–8]. In a high-speed dynamic environment, the angle measurement error caused by aero-optical effects is even as high as 400 μ rad [9,10]. Therefore, the study of aero-optical effects is of great significance for the error compensation of CNSs of hypersonic vehicles.

Aero-optical effects involve the coupling of two complex systems, which are the high-speed flow field and light field. A complex flow field generated between an optical head cover and an incoming flow causes transmission interference in an imaging detection system, causing the offsetting, jittering, blurring, and energy loss of the received image [11,12]. In recent years, many scholars have studied the measurement and distortion characteristics of aero-optical effects, mainly focusing on the analysis of flow field structure and image distortion. The high-frequency sampling structure of high-speed turbulence could be obtained via laser scattering technology [13,14]. Through a statistical analysis, it could be found that the distortion characteristics caused by aero-optical effects are closely related to different turbulence scale structures [15–20]. Vortex structures of different scales can lead to different errors in optical sensors. The scattering of small-scale vortex structures is strong, causing point blurring; large-scale vortex structures mainly cause jittering and point shift.

Although research on aero-optical effects has made some progress through experiments, research on their microscopic mechanism has stagnated [21–23]. At present, research on the mechanism of aero-optical effects mainly focuses on the change in the turbulent density field on the optical path. By analyzing the refractive index field near the turbulent boundary layer (TBL), the phase distortion caused by a high-speed flow field is studied through the ray-tracing method [24,25]. In fact, the research process only completes the analysis of the deflection effects of large-scale turbulent vortex structures on the light transmission at the macro characteristic level and cannot accurately describe the energy loss of light transmission in turbulence. Some scholars described aero-optical effects through wave theory, but the amplitude change was generally ignored, and only the simplified Maxwell equations were used for light transmission in turbulence [26,27]. This research method cannot reveal the microscopic essence of aero-optical effects.

Photon theory is a theory that describes the interaction between light and matter based on the quantization of light. As shown in Figure 1, there are three processes that may occur when photons interact with gas molecules: absorption, scattering, and stimulated radiation [28]. The essence of aero-optical effects can be classified as the interaction between photons and gas molecules in a high-speed flow field. When a laminar flow or uniform gas is taken as the research object, the microscopic effects, such as scattering and absorption, are not obvious and can be directly analyzed using geometrical optics. However, when the turbulent changes are more intense, especially at the tail of the boundary layer, the large-scale structures begin to enter the dissipation region and change into small-scale structures. At this time, the density and velocity of turbulence change dramatically, and the molecular distribution in the ground state and excited state also change. At this time, only considering the refraction effect cannot meet the analysis of the actual aero-optical effects. Therefore, the microscopic mechanism of aero-optical effects is studied based on the transmission of photons in turbulence, and a simulation method of aero-optical effects based on the microscopic mechanism (MSAO) is designed to describe the perturbation effects of the high-speed flow field on photons.

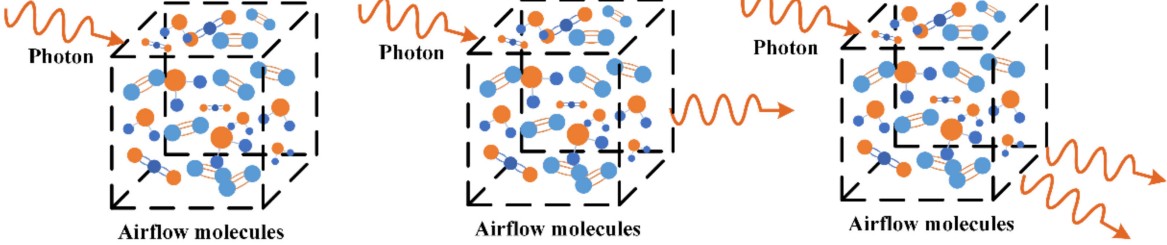

**Figure 1.** Absorption, scattering, and stimulated radiation of molecules in photon theory.

The contribution of this paper is to apply the photon transport mechanism to the microscopic analysis of aero-optical effects for the first time. Based on the interaction between photons and gas molecules in a high-speed flow field, the micro essence of the aero-optical effects is revealed from the perspective of photons. The micro simulation analysis method proposed in this paper can explain the distortion characteristics of photons in different turbulent scale structures and can also analyze the energy in the aero-optical effects.

## 2. Microscopic Mechanism of Aero-Optical Effects

The microscopic mechanism of optical distortion caused by a high-speed flow field is the interaction between photons and turbulent gas molecules in the light transmission process, as shown in Figure 2. This section studies the microscopic mechanism of the aero-optical effects based on photon transmission theory. Through analyzing the interaction between photons and turbulent molecules, the transmission law and physical characterization of photons in high-speed turbulence are obtained.

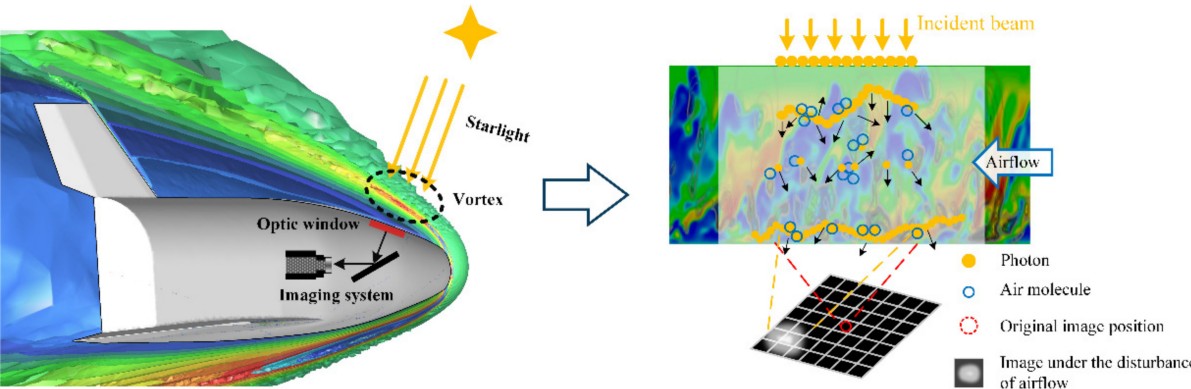

**Figure 2.** Microscopic process of photon transmission in turbulent molecules.

### 2.1. Absorption of Photons by Turbulent Molecules

Photons passing through turbulent gas molecules have a certain probability of being absorbed by gas molecules. Assuming the photons travel over a distance, $ds$, the absorption probability is $\mu_a ds$. The quantity describing the absorption process is defined as the absorption coefficient, $\mu_a(\mathbf{r}, \nu, t)$, which is a function of space, frequency, and time. The absorption of photons and gas molecules is accompanied by the polarization of electrons and the change in electronic energy states. When a photon is absorbed, the gas molecule transitions from the ground-state energy level to the excited-state energy level, as shown in Figure 3.

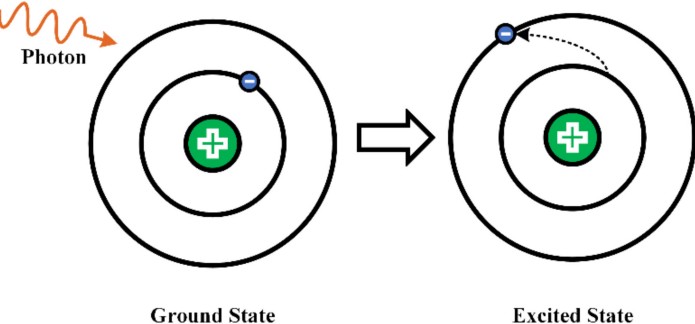

**Figure 3.** Schematic diagram of photon absorption by turbulent molecules.

Different transition modes depend on the free state of electrons after absorbing photons. Since it is very complicated to directly calculate the absorption coefficient of a single molecule with different transition modes, the complex refractive index is selected to describe the photon absorption coefficient. According to the Lorenz dispersion theory, the complex refractive index, $\widetilde{n}$, can be expressed as follows [29]:

$$\widetilde{n} = n_R + i n_I \tag{1}$$

where $n_R$ is the real part, that is, the refractive index of the medium commonly used in geometrical optics that reflects the dispersion characteristics of turbulent molecules. Additionally, $n_I$ is the imaginary part, which reflects the absorption characteristics of the molecule. The relationship between $n_I$ and $n_I$ satisfies the following:

$$\begin{cases} n_R^2 - n_I^2 = 1 + \dfrac{Ne^2}{\varepsilon_0 m_e} \dfrac{\nu_0^2 - \nu^2}{\left(\nu_0^2 - \nu^2\right)^2 + \gamma^2 \nu^2} \\[2mm] 2 n_R n_I = \dfrac{Ne^2}{\varepsilon_0 m_e} \dfrac{\gamma \nu}{\left(\nu_0^2 - \omega^2\right)^2 + \gamma^2 \nu^2} \end{cases} \tag{2}$$

where $\gamma$ is the damping coefficient, which satisfies $\gamma = e^2 \nu_0^2 / 6\pi \varepsilon_0 m_e c^3$; $\nu_0$ is the natural frequency of the polarized oscillator; $\nu$ is the photon frequency; $e$ is the electronic charge;

$m_e$ is the electronic mass; $\varepsilon_0$ is the dielectric constant of vacuum; $N$ is the molecular number density; and $c$ is the speed of light.

According to the Gladstone–Dale law, the relationship between the local gas density, $\rho$, and the macro refractive index, $n_R$, is as follows [30]:

$$n_R = 1 + K_{GD} \cdot \rho \tag{3}$$

where $K_{GD}$ is the Gladstone–Dale constant, whose value is determined by the wavelength of the beam, $\lambda$:

$$K_{GD} = 2.2244 \times 10^{-4} \left[ 1 + \left( 6.7132 \times 10^{-8} / \lambda \right)^2 \right] \text{m}^3/\text{kg} \tag{4}$$

The relationship between the absorption coefficient of the photons by turbulent molecules and the imaginary part of the complex refractive index satisfies the following:

$$\mu_a(\boldsymbol{r}, v, t) = 2 v n_{\text{I}}(\boldsymbol{r}, v, t)/c = \frac{N(\boldsymbol{r}, t)e^2}{4 m_e \varepsilon_0 c} \frac{\gamma}{(v_0 - v)^2 + (\gamma/2)^2} \tag{5}$$

Therefore, it can be found that the absorption coefficient is related to space, time, and frequency; because the distribution of turbulent gas molecules is time-varying and non-uniform, the absorption ability of turbulent molecules is also different at different times and in different regions.

### 2.2. Scattering of Photons by Turbulent Molecules

When photons pass through turbulent gas molecules, there is a certain probability that they are scattered by gas molecules. Assuming that photons travel over a distance, $ds$, the scattering probability is $\mu_s ds$. The quantity describing the scattering process is defined as the scattering coefficient, $\mu_s(\boldsymbol{r}, v, t)$.

The scattering of photons is related to the scale of the inhomogeneity of turbulent molecules. When the scale is larger than the order of the wavelength, it can be regarded as refraction, which can be explained as the synthesis of partial scattering by the Feynman quantum electrodynamics theory. The Rayleigh scattering of small-scale turbulent gas molecules is mainly considered in this paper and is caused by the optical inhomogeneity caused by the change in molecular density and dipole moment. According to the photon scattering theory, the Rayleigh scattering cross-section of a single molecule is as follows:

$$\sigma_s(\lambda) = \frac{8\pi}{3} \left[ \frac{\pi^2 \left( n_R^2 - 1 \right)^2}{N^2 \lambda^4} \right] \tag{6}$$

where $N$ is the number density of molecules and $\lambda$ is the wavelength of incident light. It can be seen that the molecular scattering cross-section is inversely proportional to the fourth power of the wavelength of the incident photon. In order to reduce the calculation amount of the scattering cross-section of a single molecule, the actual scattering coefficient is expressed as follows:

$$\mu_s(\boldsymbol{r}, v, t) = N(\boldsymbol{r}, t)\sigma_s(\lambda) = \frac{8\pi}{3c^4} \left[ \frac{\pi^2 \left( n_R(\boldsymbol{r}, v, t)^2 - 1 \right)^2 v^4}{N(\boldsymbol{r}, t)} \right] \tag{7}$$

It can be seen from Equation (7) that the scattering coefficient of turbulent molecules is also a function of space, frequency, and time. For high-speed turbulence, the increase in the degree of turbulence for gas molecules leads to the enhancement of scattering and the diffusion of photon energy.

### 2.3. Transmission Equation of Photons in Turbulent Molecules

Photons interact with gas molecules when they are transmitted in turbulence, and the number of photons changes during the transmission process. At any time $t$ of the transmission process, the position of the photons in the phase space is determined by the position coordinate, $r$, the frequency, $v$, and the direction, $\Omega$. Assuming that the distribution function of the photons is $f(\boldsymbol{r}, v, \Omega, t)$ and that $N_q$ denotes the number of photons, the number of unit photons, $dN_q$, can be expressed as follows:

$$dN_q = f(\boldsymbol{r}, v, \Omega, t)\mathrm{d}V\mathrm{d}v\mathrm{d}\Omega \tag{8}$$

where $f(\boldsymbol{r}, v, \Omega, t)$ represents the number of photons transmitting along direction $\Omega$ at time $t$, frequency $v$, and position $r$ in the unit volume, the unit frequency interval, and the unit solid angle. The photon distribution function determines the intensity, $I$, of the photon field in turbulence, which satisfies the following:

$$I(\boldsymbol{r}, v, \Omega, t) = chv f(\boldsymbol{r}, v, \Omega, t) \tag{9}$$

where $c$ is the light speed, $h$ is Planck's constant, and $I(\boldsymbol{r}, v, \Omega, t)$ represents the intensity of photons transmitting along direction $\Omega$ at time $t$, frequency $v$, and position $\boldsymbol{r}$. The Boltzmann equation of photon transmission in turbulence is as follows:

$$\frac{\partial f}{\partial t} + \nabla_r \cdot (vf) + \nabla_v \cdot (\mathbf{a}f) = \boldsymbol{Q}(\boldsymbol{r}, v, \Omega, t) \tag{10}$$

where $v$ is the photon velocity vector; $\mathbf{a}$ is the acceleration vector; and $\nabla_r$ and $\nabla_v$ are divergence operators in geometric space and velocity space, respectively. If the relativistic effect is ignored, then $\mathbf{a} = 0$, and $v = c\Omega$. $\boldsymbol{Q}(\boldsymbol{r}, v, \Omega, t)$ is the photon source term, including the interaction process between photons and turbulent gas molecules.

This paper mainly considers the absorption and scattering effects of photons. In Figure 4, local coordinate system $e_x - e_y - e_z$ is obtained via the translation of coordinate system $x - y - z$, and unit direction vector cosines $\mu, \eta$, and $\xi$ of photons are the projections of the direction vector on coordinate axes $e_x - e_y - e_z$, which can be expressed as follows:

$$\begin{cases} \mu = \sin\theta\cos\varphi \\ \eta = \sin\theta\cos\varphi \\ \xi = \cos\theta \end{cases} \tag{11}$$

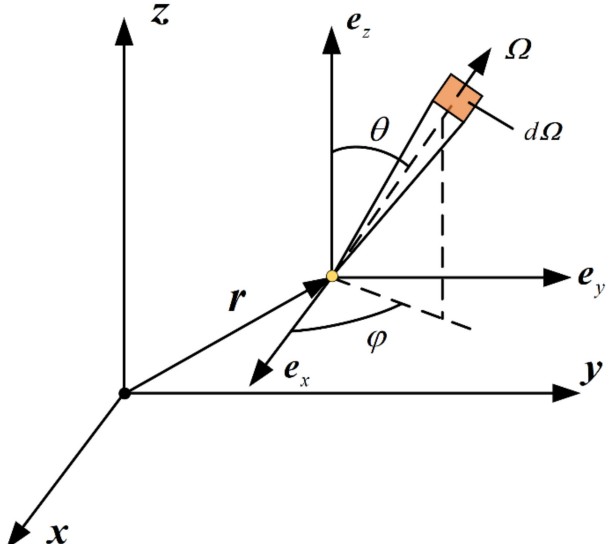

**Figure 4.** Schematic diagram of photon coordinates in a rectangular coordinate system.

According to the relationship between the photon field intensity and the photon distribution function, $I(\mathbf{r}, \nu, \mathbf{\Omega}, t) = ch\nu f(\mathbf{r}, \nu, \mathbf{\Omega}, t)$, the transmission equation of photons in turbulence can be written as the following conservation form:

$$
\begin{aligned}
&\frac{n_R}{c}\frac{\partial I(\mathbf{r},\nu,\mathbf{\Omega},t)}{\partial t} + \mu\frac{\partial I(\mathbf{r},\nu,\mathbf{\Omega},t)}{\partial x} + \eta\frac{\partial I(\mathbf{r},\nu,\mathbf{\Omega},t)}{\partial y} + \xi\frac{\partial I(\mathbf{r},\nu,\mathbf{\Omega},t)}{\partial z} \\
&+ \frac{1}{\sin\theta}\frac{\partial}{\partial\theta}\{[\xi(\mu\alpha + \eta\beta + \xi\gamma) - \gamma]I(\mathbf{r},\nu,\mathbf{\Omega},t)\} \\
&+ \frac{1}{\sin\theta}\frac{\partial}{\partial\varphi}\{[\beta\cos\varphi - \alpha\sin\varphi]I(\mathbf{r},\nu,\mathbf{\Omega},t)\} \\
&= -(\mu_a + \mu_s)I(\mathbf{r},\nu,\mathbf{\Omega},t) + \frac{\mu_s}{4\pi}\int_{4\pi} I(\mathbf{r},\nu,\mathbf{\Omega}',t)\Phi(\mathbf{\Omega},\mathbf{\Omega}')\mathrm{d}\Omega'
\end{aligned}
\tag{12}
$$

where $\Phi(\mathbf{\Omega},\mathbf{\Omega}')$ is the scattering phase function; and $\alpha$, $\beta$, and $\gamma$ can be expressed as follows:

$$
\begin{cases}
\alpha = \frac{1}{2n_R^2}\frac{\partial n_R^2}{\partial x} \\
\beta = \frac{1}{2n_R^2}\frac{\partial n_R^2}{\partial y} \\
\gamma = \frac{1}{2n_R^2}\frac{\partial n_R^2}{\partial z}
\end{cases}
\tag{13}
$$

Equation (12) is then rewritten into the form of divergence:

$$
\begin{aligned}
&\frac{n_R}{c}\frac{\partial I(\mathbf{r},\nu,\mathbf{\Omega},t)}{\partial t} + \frac{1}{2n_R^2\sin\theta}\frac{\partial}{\partial\theta}\left\{\left[I(\mathbf{r},\nu,\mathbf{\Omega},t)(\xi\mathbf{\Omega} - \mathbf{k})\cdot\nabla n_R^2\right]\right\} \\
&+ \frac{1}{2n_R^2\sin\theta}\frac{\partial}{\partial\varphi}\left\{I(\mathbf{r},\nu,\mathbf{\Omega},t)\left[\mathbf{s}\cdot\nabla n_R^2\right]\right\} + \mathbf{\Omega}\cdot\nabla I(\mathbf{r},\nu,\mathbf{\Omega},t) \\
&= -(\mu_a + \mu_s)I(\mathbf{r},\nu,\mathbf{\Omega},t) + \frac{\mu_s}{4\pi}\int_{4\pi} I(\mathbf{r},\nu,\mathbf{\Omega}',t)\Phi(\mathbf{\Omega},\mathbf{\Omega}')\mathrm{d}\Omega'
\end{aligned}
\tag{14}
$$

where $\mathbf{i}$, $\mathbf{j}$ and $\mathbf{k}$ are the unit vectors of the $\mathbf{x}$, $\mathbf{y}$ and $\mathbf{z}$ coordinate directions, respectively.

## 3. Design of a Simulation Method for Aero-Optical Effects Based on the Microscopic Mechanism

The basis of simulating aero-optical effects is the calculation of the flow field and the light field. In this paper, high-speed turbulence is calculated using large eddy simulation (LES) technology, and the physical quantity is designed based on the microscopic mechanism of photon transmission to describe the photon distribution in high-speed turbulence. The following is the design of an aero-optical effect simulation method based on the microscopic mechanism (MSAO).

The operation process of the MSAO is as follows:

Step 1: Determine the structure and flight conditions of the aircraft, conduct a large eddy simulation based on computational fluid dynamics, and obtain the turbulent flow field data under the corresponding simulation conditions;

Step 2: Extract the density data from the flow field data and calculate the absorption and scattering coefficients according to the microscopic mechanism in Section 2;

Step 3: Conduct a numerical simulation through the transmission equation of photons in turbulent molecules to obtain the transmission law of photons in turbulent molecules. The design flow chart is shown in Figure 5.

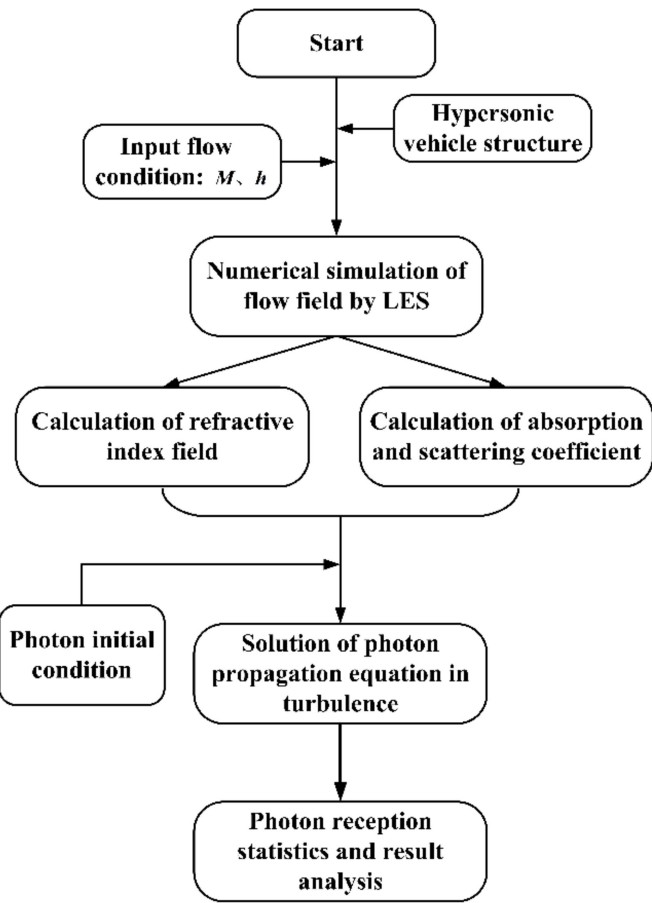

**Figure 5.** Operation process diagram of MSAO.

### 3.1. High-Speed Turbulence via LES

According to the current development of numerically simulating the flow field, it can be found that LES is a common method with which to obtain the transient structure of a hypersonic flow field [31]. In this paper, the warhead part of a hypersonic vehicle is used for flow field simulation. When the vehicle is installed with optical equipment, local surfaces are concave or open, forming concave windows. The physical structure of the flow field simulation designed in this paper is shown in Figure 6. $x_b, y_b, z_b$ and $x_w, y_w, z_w$ represent the vehicle body coordinate system and window coordinate system, respectively. The directions of $y_b$ and $y_w$ meet the right-hand coordinate system rule, and the origin of the window coordinate system is located in the center of the window. The nose angle of the vehicle is 13.5°, and the structural dimension of the optical concave window is $185 \times 125$ mm$^2$. Its optical active window size is $80 \times 80$ mm$^2$.

In this paper, commercial software Fluent 2021R2 is used to calculate the flow field. Meshing a vehicle is the first step of LES calculations. Since the flow field structure near the optical window of the vehicle is the key to analyzing aero-optical effects, the grids near the window are densified, as shown in Figure 7. The total number of grids is 6.42 million, and the thickness of the first layer of the optical window grids is 0.0002 mm.

To verify the grid independence of the model, the first layer of the model boundary layer is divided into different thicknesses, and the weighted average velocity of section $y_w - z_w$ above the window center is taken as the test standard. The coarsest grid ($5.31 \times 10^4$ *cells*), the coarse grid ($1 \times 10^5$ *cells*), the medium grid ($8.03 \times 10^5$ *cells*), the fine grid ($6.42 \times 10^6$ *cells*), and the finest grid ($5.35 \times 10^8$ *cells*) were employed to verify grid independence. The grid independence verification results are shown in Figure 8. With the increase in the number of grids, the average speed value changes less and less. Under the condition of the fine grid, the average velocity deviation is less than 0.1%. Consider-

ing the accuracy and efficiency of the calculation, the number of grids is determined as 6.42 million.

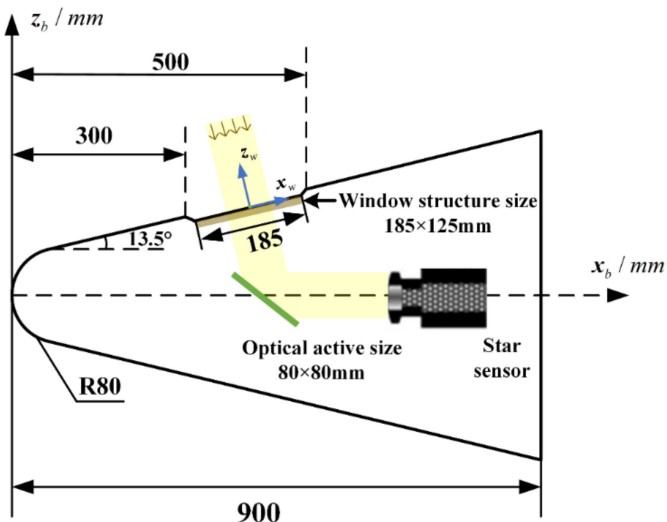

**Figure 6.** Physical structure diagram of a hypersonic vehicle.

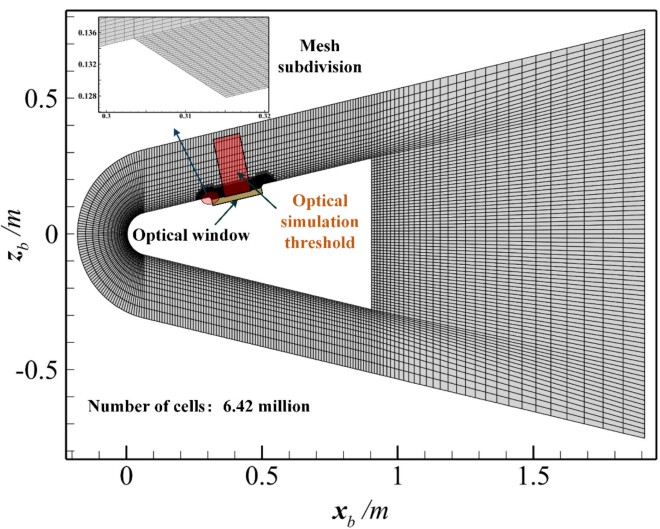

**Figure 7.** Grid distribution around the hypersonic vehicle and the optical window.

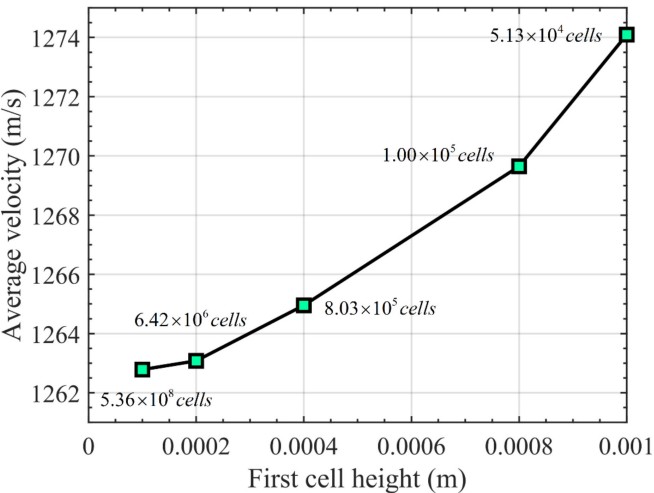

**Figure 8.** Verification of grid independence for the model.

The simulation conditions for LES are as follows: incoming flow Mach number, $Ma = 3.8$; flight altitude, $h = 20$ km; and angle of attack, $0°$. The boundary conditions in the calculation are set as follows: the outlet is set as the constant pressure outlet condition; the wall is set as a non-slip adiabatic wall; and the calculation time step is taken as $10^{-7}$ s. In order to improve the computational efficiency, a high-performance server is used to calculate the flow field. The calculated flow field results are shown in Figure 9. Compared with the schlieren images obtained from experiments in the literature [23], as shown in Figure 10, the bow shock wave, expansion shock wave, induced shock wave, and oblique shock wave are basically reproduced. Because there is still some error between the simulation and the experiment, the oblique shock wave is not particularly obvious. However, it can basically explain the correctness of the simulation.

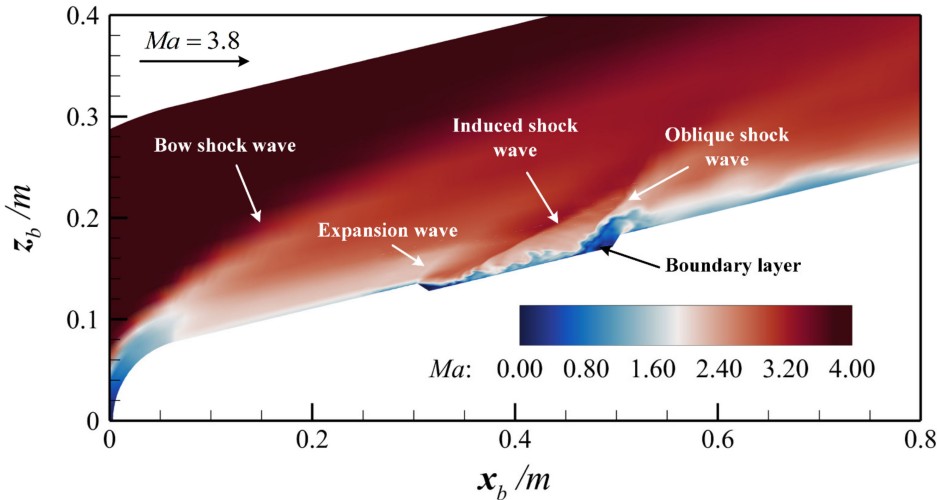

**Figure 9.** Shock wave and turbulence in high-speed flow field.

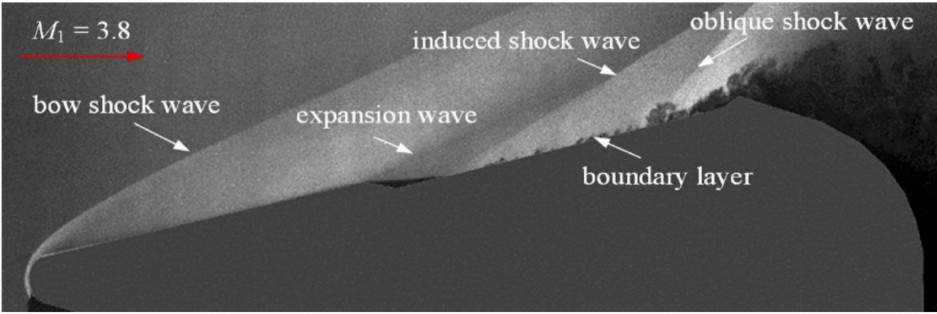

**Figure 10.** Experimental result of hypersonic turbulent flow field over the optical window [23].

### 3.2. Numerical Simulation of Photon Transmission in Turbulence

In this paper, the MSAO is carried out based on the microscopic mechanism of photon transmission, whose design is based on the following assumptions:

- The photon transmission in turbulent molecules does not consider stimulated radiation;
- The interaction between light and turbulent molecules is described by the absorption coefficient and the scattering coefficient;
- The polarization and interference of photons are ignored.

The transmission equation of photons is a differential equation, and the variables are space coordinates and time coordinates. Since the calculation of time variables would lead to a sharp increase in the calculation amount, as well as the fact that the photon movement speed is far higher than the change speed of the flow field, it can be considered that the turbulent structure is static during the interaction between photons and turbulent

molecules. Therefore, this paper only considers the simulation of photon transmission in a certain turbulent state. Therefore, Equation (14) can be written as follows:

$$
\begin{aligned}
&\mathbf{\Omega} \cdot \nabla I(\mathbf{r}, \nu, \mathbf{\Omega}) + \frac{1}{2n_R^2 \sin\theta} \frac{\partial}{\partial\theta} \left\{ \left[ I(\mathbf{r}, \nu, \mathbf{\Omega})(\xi\mathbf{\Omega} - \mathbf{k}) \cdot \nabla n_R^2 \right] \right\} \\
&+ \frac{1}{2n_R^2 \sin\theta} \frac{\partial}{\partial\varphi} \left\{ I(\mathbf{r}, \nu, \mathbf{\Omega}) \left[ \mathbf{s} \cdot \nabla n_R^2 \right] \right\} \\
&= -(\mu_a + \mu_s) I(\mathbf{r}, \nu, \mathbf{\Omega}) + \frac{\mu_s}{4\pi} \int_{4\pi} I(\mathbf{r}, \nu, \mathbf{\Omega}') \Phi(\mathbf{\Omega}, \mathbf{\Omega}') \mathrm{d}\mathbf{\Omega}'
\end{aligned}
\tag{15}
$$

In this paper, the discrete coordinate method is used to solve the differential equation. The equation contains not only the spatial differential term but also the angle integral term. Therefore, the space and angle need to be discretized separately. Firstly, the space computing domain is divided into $N_V$ volume units, $\Delta V_p$, and the solid angle is divided into $N_\theta \times N_\varphi$ parts via the method of piecewise constant angle (PCA) quadrature. Therefore, the above equation can be written in the form of discrete coordinates as follows:

$$
\begin{aligned}
&\mathbf{\Omega}^{m,n} \cdot \nabla I(\mathbf{r}, \nu, \mathbf{\Omega}^{m,n}) + \frac{1}{2n_R^2 \sin\theta^m} \left\{ \frac{\partial}{\partial\theta} \left[ I(\mathbf{r}, \nu, \mathbf{\Omega})(\xi\mathbf{\Omega} - \mathbf{k}) \cdot \nabla n_R^2 \right] \right\}_{\mathbf{\Omega}=\mathbf{\Omega}^{m,n}} \\
&+ \frac{1}{2n_R^2 \sin\theta^m} \left\{ \frac{\partial}{\partial\varphi} \left[ I(\mathbf{r}, \nu, \mathbf{\Omega}) \left[ \mathbf{s} \cdot \nabla n_R^2 \right] \right] \right\}_{\mathbf{\Omega}=\mathbf{\Omega}^{m,n}} \\
&= -(\mu_a + \mu_s) I(\mathbf{r}, \nu, \mathbf{\Omega}^{m,n}) + \frac{\mu_s}{4\pi} \sum_{m'=1}^{N_\theta} \sum_{n'=1}^{N_\varphi} I(\mathbf{r}, \nu, \mathbf{\Omega}^{m',n'}) \Phi\left(\mathbf{\Omega}^{m',n'}, \mathbf{\Omega}^{m,n}\right) \omega_\theta^{m'} \omega_\varphi^{n'}
\end{aligned}
\tag{16}
$$

where the discrete zenith angle and circumference angle are shown as follows:

$$
\begin{aligned}
\theta^m &= (m - 1/2)\Delta\theta, & m = 1, 2, \cdots, N_\theta \\
\varphi^n &= (n - 1/2)\Delta\varphi, & n = 1, 2, \cdots, N_\varphi
\end{aligned}
\tag{17}
$$

where $\Delta\theta = \pi/N_\theta, \Delta\varphi = 2\pi/N_\varphi$. For each discrete direction, the corresponding weights are as follows:

$$
\begin{aligned}
\omega_\theta^m &= \int_{\theta^{m-1/2}}^{\theta^{m+1/2}} \sin\theta \mathrm{d}\theta = \cos\theta^{m-1/2} - \cos\theta^{m+1/2} \\
\omega_\varphi^n &= \int_{\varphi^{n-1/2}}^{\varphi^{n+1/2}} \mathrm{d}\varphi = \varphi^{n+1/2} - \varphi^{n-1/2}
\end{aligned}
\tag{18}
$$

The derivatives of the zenith angle and the circumference angle require finite difference approximation, and the angle distribution term can be written as follows:

$$
\begin{aligned}
&\frac{1}{2n_R^2 \sin\theta^m} \left\{ \frac{\partial}{\partial\theta} \left[ I(\mathbf{r}, \nu, \mathbf{\Omega})(\xi\mathbf{\Omega} - \mathbf{k}) \cdot \nabla n_R^2 \right] \right\}_{\mathbf{\Omega}=\mathbf{\Omega}^{m,n}} \\
&\approx \frac{\chi_\theta^{m+1/2,n} I(\mathbf{r}, \nu, \mathbf{\Omega}^{m+1/2,n}) - \chi_\theta^{m-1/2,n} I(\mathbf{r}, \nu, \mathbf{\Omega}^{m-1/2,n})}{\omega_\theta^m}
\end{aligned}
\tag{19}
$$

$$
\begin{aligned}
&\frac{1}{2n_R^2 \sin\theta^m} \left\{ \frac{\partial}{\partial\varphi} \left[ I(\mathbf{r}, \nu, \mathbf{\Omega}) \left[ \mathbf{s} \cdot \nabla n_R^2 \right] \right] \right\}_{\mathbf{\Omega}=\mathbf{\Omega}^{m,n}} \\
&\approx \frac{\chi_\varphi^{m,n+1/2} I(\mathbf{r}, \nu, \mathbf{\Omega}^{m,n+1/2}) - \chi_\varphi^{m,n-1/2} I(\mathbf{r}, \nu, \mathbf{\Omega}^{m,n-1/2})}{\omega_\varphi^n}
\end{aligned}
\tag{20}
$$

Therefore, the discrete coordinate equation of photon propagation in turbulent molecules is as follows:

$$
\begin{aligned}
&\mathbf{\Omega}^{m,n} \cdot \nabla I(\mathbf{r}, \nu, \mathbf{\Omega}^{m,n}) + \frac{\chi_\theta^{m+1/2,n} I(\mathbf{r}, \nu, \mathbf{\Omega}^{m+1/2,n}) - \chi_\theta^{m-1/2,n} I(\mathbf{r}, \nu, \mathbf{\Omega}^{m-1/2,n})}{\omega_\theta^m} \\
&+ \frac{\chi_\varphi^{m,n+1/2} I(\mathbf{r}, \nu, \mathbf{\Omega}^{m,n+1/2}) - \chi_\varphi^{m,n-1/2} I(\mathbf{r}, \nu, \mathbf{\Omega}^{m,n-1/2})}{\omega_\varphi^n} \\
&= -(\mu_a + \mu_s) I(\mathbf{r}, \nu, \mathbf{\Omega}^{m,n}) + \frac{\mu_s}{4\pi} \sum_{m'=1}^{N_\theta} \sum_{n'=1}^{N_\varphi} I(\mathbf{r}, \nu, \mathbf{\Omega}^{m',n'}) \Phi\left(\mathbf{\Omega}^{m',n'}, \mathbf{\Omega}^{m,n}\right) \omega_\theta^{m'} \omega_\varphi^{n'}
\end{aligned}
\tag{21}
$$

The following discrete coordinate forms can be obtained by integrating Equation (21) on the volume element:

$$
\begin{aligned}
&\sum_{i=1}^{N_P} (\boldsymbol{\Omega}^{m,n} \cdot \boldsymbol{e}_i) A_i I_i(\boldsymbol{r}, \nu, \boldsymbol{\Omega}^{m,n}) \\
&+ \frac{\overline{\chi}_\theta^{m+1/2,n} I(\boldsymbol{r},\nu,\boldsymbol{\Omega}^{m+1/2,n}) - \overline{\chi}_\theta^{m-1/2,n} I(\boldsymbol{r},\nu,\boldsymbol{\Omega}^{m-1/2,n})}{\omega_\theta^m} \\
&+ \frac{\overline{\chi}_\varphi^{m,n+1/2} I(\boldsymbol{r},\nu,\boldsymbol{\Omega}^{m,n+1/2}) - \overline{\chi}_\varphi^{m,n-1/2} I(\boldsymbol{r},\nu,\boldsymbol{\Omega}^{m,n-1/2})}{\omega_\varphi^n} \\
&= -\Delta V_P (\mu_a + \mu_s) I(\boldsymbol{r}, \nu, \boldsymbol{\Omega}^{m,n}) \\
&+ \frac{\Delta V_P \kappa_s}{4\pi} \sum_{m'=1}^{N_\theta} \sum_{n'=1}^{N_\varphi} I(\boldsymbol{r}, \nu, \boldsymbol{\Omega}^{m',n'}) \Phi\left(\boldsymbol{\Omega}^{m',n'}, \boldsymbol{\Omega}^{m,n}\right) \omega_\theta^{m'} \omega_\varphi^{n'}
\end{aligned}
\tag{22}
$$

$$
\overline{\chi}_\theta^{m+1/2,n} - \overline{\chi}_\theta^{m-1/2,n} = \frac{w_\theta^m}{2 n_{R,P}^2 \sin\theta^m} \sum_{i=1}^{N_P} \left\{ \left[ \left( \frac{\partial(\xi\boldsymbol{\Omega})}{\partial\theta} \right)_{\boldsymbol{\Omega}=\boldsymbol{\Omega}^{m,n}} \cdot \boldsymbol{e}_i \right] A_i n_{R,i}^2 \right\}
\tag{23}
$$

$$
\overline{\chi}_\theta^{1/2,n} = \overline{\chi}_\theta^{N_\theta+1/2,n} = 0
\tag{24}
$$

$$
\overline{\chi}_\varphi^{m,n+1/2} - \overline{\chi}_\varphi^{m,n-1/2} = \frac{\omega_\varphi^n}{2 n_{R,P}^2 \sin\theta^m} \sum_{i=1}^{N_P} \left[ \left( \frac{\partial\boldsymbol{s}}{\partial\varphi} \right)_{\boldsymbol{\Omega}=\boldsymbol{\Omega}^{m,n}} \cdot \boldsymbol{e}_i \right] A_i n_{R,i}^2
\tag{25}
$$

$$
\overline{\chi}_\varphi^{m,1/2} = \overline{\chi}_\varphi^{m,N_\theta+1/2} = \frac{1}{2 n_{R,P}^2 \sin\theta^m} \sum_{i=1}^{N_P} (\boldsymbol{j} \cdot \boldsymbol{e}_i) A_i n_{R,i}^2
\tag{26}
$$

where $N_P$ is the number of surfaces of unit $P$; $A_i$ is the area of the $i$-th surface; $\boldsymbol{e}_i$ and $n_{R,i}$ are, respectively, the normal vector and refractive index outside the unit of the $i$-th surface; $\Delta V_p$ is the volume of unit $P$; and $n_{R,P}$ represents the refractive index of the center of unit $P$. In order to make the integral equation closed, boundary conditions need to be set. Here, the strength value of the downstream surface is selected to be equal to the strength value of the upstream center, meeting the following:

$$
\begin{aligned}
&(\boldsymbol{\Omega}^{m,n} \cdot \boldsymbol{e}_i) A_i I_i(\boldsymbol{r}, \nu, \boldsymbol{\Omega}^{m,n}) = \frac{n_{R,i}^2}{n_{R,P}^2} \max[(\boldsymbol{\Omega}^{m,n} \cdot \boldsymbol{e}_i) A_i, 0] I_P(\boldsymbol{r}, \nu, \boldsymbol{\Omega}^{m,n}) \\
&- \frac{n_{R,i}^2}{n_{R,P_i}^2} \max[-(\boldsymbol{\Omega}^{m,n} \cdot \boldsymbol{e}_i) A_i, 0] I_{P_i}(\boldsymbol{r}, \nu, \boldsymbol{\Omega}^{m,n})
\end{aligned}
\tag{27}
$$

$$
\begin{aligned}
&\overline{\chi}_\theta^{m+1/2,n} I_P(\boldsymbol{r}, \nu, \boldsymbol{\Omega}^{m+1/2,n}) = \max\left(\overline{\chi}_\theta^{m+1/2,n}, 0\right) I_P(\boldsymbol{r}, \nu, \boldsymbol{\Omega}^{m,n}) \\
&- \max\left(-\overline{\chi}_\theta^{m+1/2,n}, 0\right) I_P(\boldsymbol{r}, \nu, \boldsymbol{\Omega}^{m+1,n})
\end{aligned}
\tag{28}
$$

$$
\begin{aligned}
&\overline{\chi}_\theta^{m-1/2,n} I_P(\boldsymbol{r}, \nu, \boldsymbol{\Omega}^{m-1/2,n}) = \max\left(\overline{\chi}_\theta^{m-1/2,n}, 0\right) I_P(\boldsymbol{r}, \nu, \boldsymbol{\Omega}^{m-1,n}) \\
&- \max\left(-\overline{\chi}_\theta^{m-1/2,n}, 0\right) I_P(\boldsymbol{r}, \nu, \boldsymbol{\Omega}^{m,n})
\end{aligned}
\tag{29}
$$

$$
\begin{aligned}
&\overline{\chi}_\varphi^{m,n+1/2} I_P(\boldsymbol{r}, \nu, \boldsymbol{\Omega}^{m,n+1/2}) = \max\left(\overline{\chi}_\varphi^{m,n+1/2}, 0\right) I_P(\boldsymbol{r}, \nu, \boldsymbol{\Omega}^{m,n}) \\
&- \max\left(-\overline{\chi}_\varphi^{m,n+1/2}, 0\right) I_P(\boldsymbol{r}, \nu, \boldsymbol{\Omega}^{m,n+1})
\end{aligned}
\tag{30}
$$

$$
\begin{aligned}
&\overline{\chi}_\varphi^{m,n-1/2} I_P(\boldsymbol{r}, \nu, \boldsymbol{\Omega}^{m,n-1/2}) = \max\left(\overline{\chi}_\varphi^{m,n-1/2}, 0\right) I_P(\boldsymbol{r}, \nu, \boldsymbol{\Omega}^{m,n-1}) \\
&- \max\left(-\overline{\chi}_\varphi^{m,n-1/2}, 0\right) I_P(\boldsymbol{r}, \nu, \boldsymbol{\Omega}^{m,n})
\end{aligned}
\tag{31}
$$

where $P_i$ is the adjacent unit sharing the $i$-th surface with unit $P$ and $I_{P_i}(\boldsymbol{r}, \nu, \boldsymbol{\Omega}^{m,n})$ represents the photon intensity of unit $P_i$. The above process is the spatial coordinate discretization in one direction, and the other directions are the same. In this way, a discrete equation group is formed. The photon intensity is solved by solving and iterating the discrete equation.

### 3.3. Physical Description of Aero-Optical Effects Based on the MSAO

In terms of the physical quantities of the traditional aero-optical effects, the optical path difference (OPD), obtained using the geometric ray-tracing method, the point spread function (PSF), and the Strehl ratio (SR) of the received image are generally used to describe

aero-optical distortion. In this section, based on photon theory, the distortion physical quantity of microscopic photons in turbulent molecules is established to describe the perturbation process caused by aero-optical effects.

### 3.3.1. Photon Energy Divergence (PED)

The PED takes a certain transmission plane as an object. Assuming that the vector, $r$, of photons is on a certain reception plane, $\Lambda$, and that the scattering does not change the frequency of photons during transmission (without considering Raman scattering); the PED at a certain plane, $\Lambda$, is the concentration of the distribution of photons on that plane, which can be expressed as follows:

$$PED(\Lambda, \nu, t) = \sum_{r \in \Lambda} |r - r_0| \frac{f(r, \nu, \mathbf{\Omega}, t)}{\sum\limits_{r \in \Lambda} f(r, \nu, \mathbf{\Omega}, t)} \tag{32}$$

where $r_0$ is the vector diameter on the receiving plane in the case of photon linear transmission. Therefore, the higher the value of the PED is, the greater the dispersion of photon transmission is, and the stronger the scattering effect encountered in the process of photon transmission is. Compared with the traditional PSF, which only focuses on the light field distribution of the optical system for the point light source, the PED can more conveniently directly capture the concentration of photon distribution on a receiving plane.

### 3.3.2. Energy Dissipation Ratio (EDR)

Unlike the description of the PED, which mainly focuses on scattering, the EDR tends to describe the absorption physically. For a certain transmission plane, $\Lambda$, the EDR represents the loss ratio between the total energy of photons on the plane and the energy of photons of the initial light source:

$$EDR(\Lambda, \nu, t) = \sum_{r \in \Lambda} \frac{I_0 - ch\nu f(r, \nu, \mathbf{\Omega}, t)}{I_0} \tag{33}$$

where $I_0$ is the intensity of the initial light source. Therefore, the larger the EDR value is, the greater the energy dissipation in the photon transmission process is, and the stronger the absorption effect of turbulent molecules on photons is. Here, compared with the traditional SR, which only compares the peak light intensity, the EDR can more comprehensively describe the loss of overall photon energy.

### 3.3.3. Photon Deflection Angle (PDA)

For a certain transmission plane, $\Lambda$, the PDA during photon transmission is defined as the angle between the transmission direction and the initial direction of photons for any vector $r$ on plane $\Lambda$; the average photon deflection angle, $\overline{PDA}$, is defined as the sum of the product of the weight of the photon number for any vector $r$ on plane $\Lambda$ and the deflection angle:

$$PDA(r, \nu, t) = \langle \mathbf{\Omega}(r, \nu, t) - \mathbf{\Omega}_0 \rangle, r \in \Lambda \tag{34}$$

$$\overline{PDA}(\Lambda, \nu, t) = \sum_{r \in \Lambda} PDA(r, \nu, t) \frac{f(r, \nu, \mathbf{\Omega}, t)}{\sum\limits_{r \in \Lambda} f(r, \nu, \mathbf{\Omega}, t)} \tag{35}$$

where $\mathbf{\Omega}_0$ is the direction vector of the initial light source. Therefore, the larger the value of the PDA is, the greater the phase distortion of photons in the transmission process is. Compared with the traditional physical description using the OPD to reflect the phase distortion, the description of the PDA is more intuitive and more accurate than the traditional method of obtaining the offset angle through centroid extraction.

## 4. Results and Discussion

In this section, the MSAO designed in Section 3 is simulated and verified to analyze the aero-optical effects of photon propagation in high-speed turbulent molecules. The optical window size of the star sensor is $80 \times 80$ mm$^2$, with an angle of view of $8°$. In order to fully simulate the photons that can reach the optical window and different flow field areas, a rectangular flow field of $120 \times 120 \times 200$ mm$^3$ is selected as the optical simulation threshold in the window coordinate system. As shown in Figure 11, the origin of the window coordinate system is located in the center of the window.

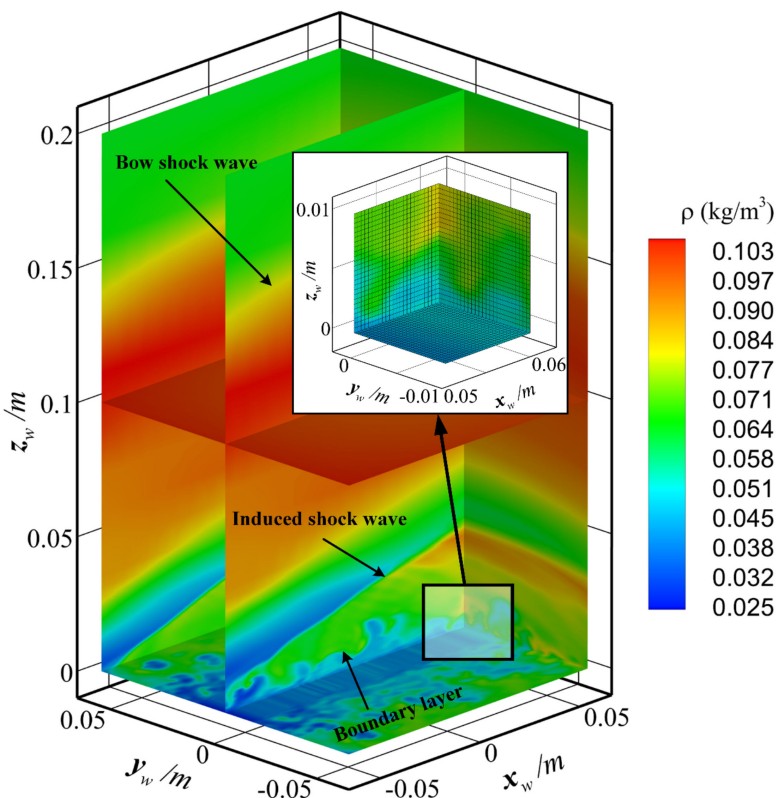

**Figure 11.** Optical simulation threshold above the window and its spatial structure division.

When using the discrete coordinate method to solve it, the whole photon calculation domain is divided into $N_x \times N_y \times N_z = 300 \times 300 \times 500$ equal parts, and the circumference angle is evenly divided into $N_\theta \times N_\varphi = 360 \times 360$ angle units. The initial condition of the light source is set as a point light source with Gaussian distribution at the center of the intercepted flow field. The initial incidence angle is 90 degrees; the wavelength is 572 nm; and the number of photons is $1 \times 10^8$.

In order to verify the correct transmission of photons in turbulent molecules, we first simulate the transmission of point light sources in a uniform medium and replace the flow field in Figure 11 with a uniform medium with a refractive index of 1, regardless of absorption and scattering. The photon reception intensity at the optical window is shown in Figure 12.

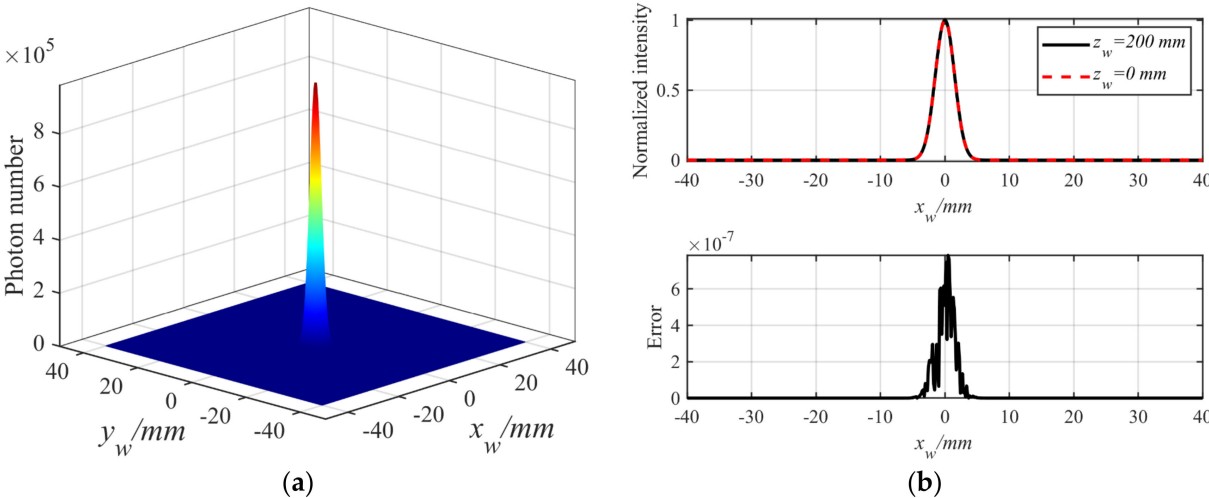

**Figure 12.** (**a**) Photon reception intensity in the ideal optical system; (**b**) optical calculation error in the ideal optical system.

Since the photon passes through a homogeneous medium with a refractive index of 1 and absorption nor scattering is considered, the photon intensity at the top and bottom of the simulation threshold should be the same. The error curve in Figure 12b can prove the correctness of the photon transmission calculation process. The photon simulation threshold in Figure 11 is then simulated, and the photon distribution at the optical window is calculated, as shown in Figure 13.

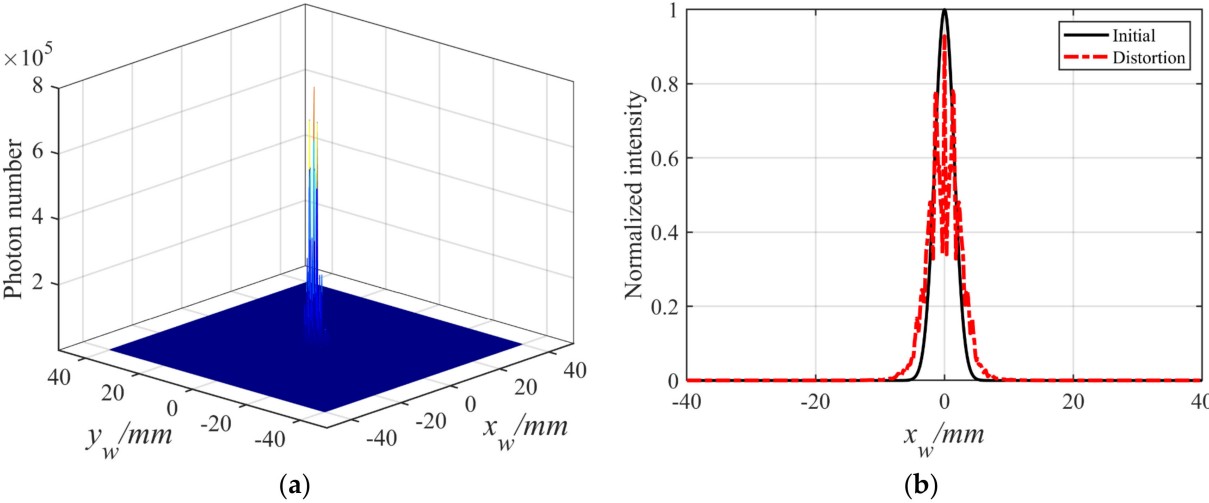

**Figure 13.** (**a**) Photon reception intensity at the optical window; (**b**) comparison of normalized photon reception intensity.

It can be seen from Figure 13 that the photon distribution is distorted, and the photon intensity is degraded, with a Strehl ratio (SR) of 0.9283. In fact, the SR can only represent the degradation of the maximum light intensity, which is caused by optical differences, without considering the energy loss in actual transmission. Therefore, the physical indicators of the photon-receiving surface at different heights are calculated. The simulation results of different receiving heights can be obtained by solving the photon transmission equation. Assuming that the bottom of the optical window satisfies $z_w = 0$, the interception ranges of different receiving planes are $z_w = 0 : 200$ mm, $x_w \in (-40$ mm$, 40$ mm$)$, and $y_w \in (-40$ mm$, 40$ mm$)$ in the window coordinate system. The PED and EDR at different heights are as shown in Figure 14.

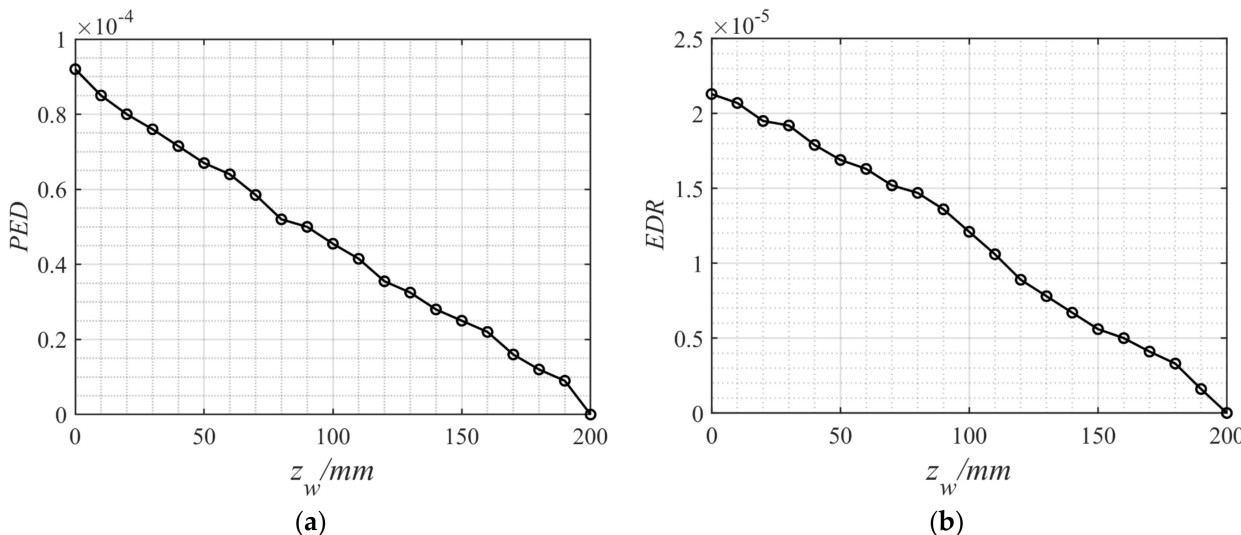

**Figure 14.** (**a**) PED for receiving planes of different heights; (**b**) EDR for receiving planes of different heights.

Figure 14 shows that the PED and EDR of photons on the receiving plane closer to the optical window are larger, which indicates that the absorption and scattering effect of turbulent molecules on photons is stronger with the increase in the transmission distance. Moreover, the intensity of PED and EDR changes approximately linearly with the increase in the thickness of the photon simulation threshold. Although the EDR value is smaller, the loss ratios of photon energy at different positions are different, resulting in the identification of the center of mass being affected when calculating the final offset angle with the traditional method.

The PED and EDR reflect the energy characteristics of photon transmission. The traditional method uses the optical path difference (OPD) to describe the phase distortion. In this paper, when receiving photons at the optical window, the phase distortion of photons can be reflected by directly calculating the PDA, as shown in Figure 15.

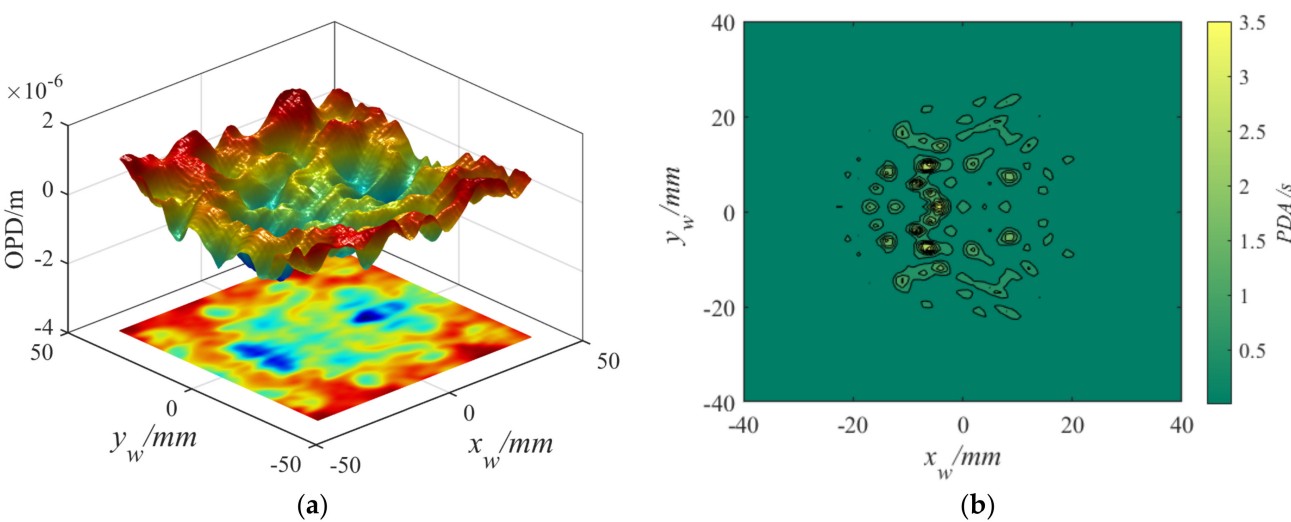

**Figure 15.** (**a**) OPD distribution according to the traditional method; (**b**) PDA distribution at optical window reception.

It is obvious that although the PDA has a certain concentration, there are certain differences in the PDAs at different positions due to the uneven distribution of molecules in high-speed turbulence. In the traditional aero-optical effects analysis process, the OPD is used to solve the optical transfer function, and the center of mass extraction method is

generally used to calculate the offset angle (COA). In order to verify the correctness of the simulation method, $\overline{PDA}$ and the center of mass extraction method are separately used to calculate the final offset angle, as shown in Figure 16. In addition, Figure 11 shows that there is a shock wave structure near $z_w = 50$ mm and $z_w = 150$ mm. It can be found from Figure 16 that the change in offset angle is relatively greater near $z_w = 50$ mm and $z_w = 150$ mm, which can explain the deflection effect of the shock wave structure on light.

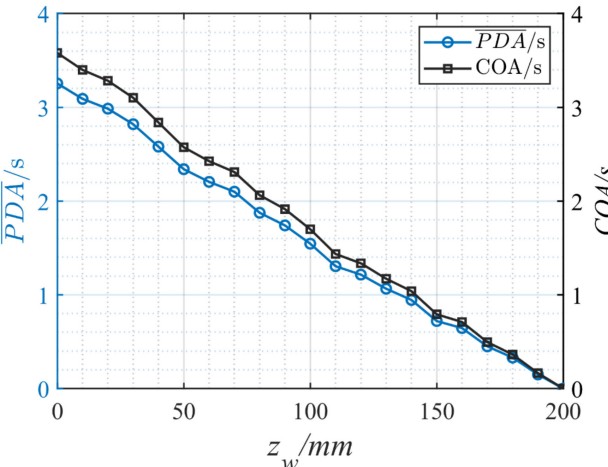

**Figure 16.** Comparison of two methods for calculating the final offset angle.

In fact, the $\overline{PDA}$ calculated using the photon transmission method has the same trend as the COA obtained using the traditional centroid extraction method, and the offset angle increases with the increase in the transmission distance. It can verify the correctness of the photon transmission method. However, the average OPA is slightly smaller than the COA in the calculation process, because the photon transmission simulation method considers not only the point with the largest energy but also the comprehensive effect of all energy offset points; the actual energy loss is also considered, which should be more in line with the actual situation than the direct centroid extraction method. The simulation results show that the offset angle of the optical window surface would be overestimated by 5.56% using traditional methods.

## 5. Conclusions

In this paper, a new method for analyzing aero-optical effects based on the interaction between photons and gas molecules is proposed. It is the first time that the photon transmission mechanism is applied to the microscopic analysis of aero-optical effects, and the microscopic essence of aero-optical effects is revealed from the perspective of photons. Because it is difficult to analyze the energy dissipation in the transmission process via the existing aero-optical simulation methods, this paper designs a simulation analysis method to explain the optical distortion and energy dissipation phenomenon caused by the aero-optical effects at the micro level that can explain the distortion characteristics of photons in different turbulent scale structures and can analyze the energy in the aero-optical effects, making up for the limitations of the traditional geometric optical method. By comparing the optical distortion parameters designed from the perspective of photons with the physical quantities of traditional aero-optical effects, the effectiveness of a micro analysis on the macro scale is verified, which could promote a better application of optical sensors in hypersonic vehicles in the future.

**Author Contributions:** Conceptualization, B.Y. and H.Y.; methodology, H.Y.; software, C.L.; validation, H.Y., X.W. and Z.F.; formal analysis, J.M.; investigation, H.Y.; resources, C.L.; data curation, H.Y.; writing—original draft preparation, H.Y.; writing—review and editing, B.Y.; visualization, B.Y.; supervision, B.Y.; project administration, H.Y.; funding acquisition, B.Y. All authors have read and agreed to the published version of the manuscript.

**Funding:** This research study was funded by Science and Technology on Space Intelligent Control Laboratory of China (No. ZDSYS-2018-03), National Natural Science Foundation of China (No. 61973018), and Civil Aerospace Technology Pre-Research Project of China (No. D040301).

**Conflicts of Interest:** The authors declare no conflict of interest.

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
