# Peer review of "A New Method for Analyzing the Aero-Optical Effects of Hypersonic Vehicles Based on a Microscopic Mechanism"

_aerospace, doi:10.3390/aerospace9100618_

Round 1

Reviewer 1 Report

Rewrite the part keywords, keyword is generally one or two words, not a sentence. Try to use e.g. IEEE taxonomy (https://www.ieee.org/content/dam/ieee-org/ieee/web/org/pubs/ieee-taxonomy.pdf),

Expand the use of CNS systems on the board of hypersonic vehicles, e.g. influence of the accuracy by aero-optical effects.

Rows 89-93 presenting the contain of the paper are useless, exclude them,

Figure 3 should have better resolution.

Equations are the parts of sentences. Use "," or "."

Rows 201, 202 (Fig. 4). Why this physical structure? Is this the part of real aircraft or aircraft concept?

Frequent repetition "it can be found" rows 133,294,305,314. Substitute by synonym.

Expand the conclusion by adding the results.

Reviewer 2 Report

Authors have claimed that this case would be the first attempt of microscopic analysis of aero-optics computation to predict macroscopic aero-optical behavior. However, it seems likely that current manuscript is lack of detailed description of used CFD code whether in-house developed code or commercially available code, evidences of grid independency test, description on grid numbers used,  CFD validational example, detailed description of optics solution code again whether in-house code or commercially available code, information of grid system for optics computation, optics computation validational example.

To proceed the CFD simulation, it is necessary to describe CFD code details concisely with grid independency test evidences, and validational example compared to the experimental cases (as this hemispherical head cone geometry is quite common, ther would be quite many excellent experimental results available in archival journal papers with Schlieren flow visualization that authors can compare shock structures of blunt normal shock in front of hemispherical head, and oblique shock laid on the conicl main body.), Also, it is necessary to show the grid numbers how many was used, being what would be the finest grid dimensions, as LES need quite large numbers of grid with fine grid system. There were long stories of optical treatment and equations, however, there is nothing for the optics computational schemes how to solve those nonlinear partial differential equations. There is no details of optical calculational domain. Maybe, it would be better to add Optical Computational Domain and Computational Grid System overlayed on the the CFD domain shown in Figure 5. Also, it would be better, easily readable, if all the dimensional information in Figure 4 merged to Figure 5 together with optical computational domain. Authors said the Optical Domain was 50X50X75mm parallelepipe, however, the window size seems like 300X300mm, so where is the optical domain in this window ? Also it seems like the Optical Domain height 75mm is below the oblique shock wave. Then, current optical calculation is very limited not including wave deflection behavior due to the shock wave.

So, in overall sense, Figure 5 should draw shock structures and Optical Domain overlayed onto the CFD grid systems. Of course, authors need to compute all progress of incident light experiencing natural turbulence of free stream flow, shock wave, and inner layer turbulent flows to demonstrate exellency and validity of current new aero-optical computational trial.

There were 3 axis systems of X, Y, Z, coordinates, together with greek symbols of myu, eta, zetta coordinates, and position vector r, direction Omega(Capital), plane Gamma(Capital). Audience might not be easy to follow these coordinate systems and coordinate transformations, and relationship between those 3 coordinate systems. It may be better and easier to catch up authors explanation, if these 3 coordinates systems are layed on to One drawing, maybe re-drawn Figure 5.

As the optical domain was limited to height of 75mm, it seems likely that only absorption might be computed, so the results in Figure 9 and 11 showed that any behavior would be linearly varying according to the distance of incident light travels, or it might draw the consequence that any optical behavior is just linear with distance, or if one may suggest an appropriate absoption/unit lentgth, then it would be easily predict current result dependig on distance, for example, if we take one liner slope of figure, then everything can be arithmetically calculated. It seems like that this consequence comes from the Optical Domain not including shock wave, or too short height of domain.

Minor comments

1) Several symbols need to be identified,defined or explained.  such as chv , omega (lower case character)

2) In expressions, i) G-D law == should be Glastone-Dale law, ii) gas density and .. index  is   == should be gas density [rho] and .. index nR is   iii) as same as the sequence in the function independent variables, main text should be ritten as " space, frequency, and time. 

Round 2

Reviewer 2 Report

Authors have quite greatly improved previous manuscript to be well shaped, however, there were still several points to be improved.

1) Authors have demonstrated CFD grid independency test, however, just noticed that "The grid independence verification results are shown in Figure 8. It can be seen from Figure 8 that the number of grids has little influence on the average speed value. " To be clearly stated, for grid independency test, most coarse grid (5.13X104 cell), moderate coarse grid (1.00X105 cell), coarse grid (8.03X105 cell), medium grid (6.42X106 cell), most fine grid (5.36X108 cell) were employed to get clear differences ...  and only 0.1% deviation of mean velocity assuring accuracy ...

2) General flow simulation was demonstrated, however, again just noticed  "Compared with the schlieren images obtained from experiments in the literature [23], the correctness of the turbulence calculation is verified." Actually, nobody can witness current simulation was correct or not as there was no comparison. It would be much strong influence to readers, if authors add Schlieren picture of ref [23] compared with the simulation result of Figure 9. However, simulation case was for Mach 3.0 while experimental case was for Mach 3.8, and at glance obique shock structure seems not correctly reproduced.

3) To overlay the optical grid onto the CFD grid would be immediately well understood by readers, so stongly recommend a different color optical calculation grid should be added by overlaying onto the CFD grid in Figure 7 as same scheme as in Figure 6 of ref [23].

4) Still current optical calculation might not well demonstrate the influence of oblique shock wave in Figure 14 and 15. Maybe, the presentation of Figure 14 as same style as Figure 12 would be crisp clearly shown that aero-optically influenced beams would be deflected, dipersed, blurred,intensity-weakened. nd, maybe authors also can demonstrate the beam intensity degradation if compared the O-th order intensity  of influenced beam against incident beam in Figure 12, again maybe Strehl Ratio (SR) and/or Optical Path Difference (OPD) would be derived.

5) Still certain explanation of connection between microscopic treatment and macroscopic observation seems lack in this article. However, only macroscopic aero-optical behavior could be observable and measurable though ...

6) Minor typo ; legend in Figure 'Eoblique shock wave' = should be  'Oblique shock wave'.

7) 22 references of total 31 references were originated from China only, and reference survey might be rather limited or biased, so it would be required to search more relevant references from all the global world, such as Jumper's group.
